# Genetic Screening of a Large Panel of Genes Associated with Cardiac Disease in a Spanish Heart Transplanted Cohort

Elías Cuesta-Llavona [1,2,3], Rebeca Lorca [2,3,4,5], Beatriz Díaz-Molina [2,4], José L. Lambert-Rodríguez [4], Julián R. Reguero [2,4,5], Sara Iglesias [1], Belén Alonso [1], Alejandro Junco-Vicente [4], Vanesa Alonso [4], Eliecer Coto [1,2,3,5] and Juan Gómez [1,2,3,5,6,*]

1   Genetica Molecular, HUCA, 33011 Oviedo, Spain; eliascllavona@gmail.com (E.C.-L.); saraiglesiasalvarez@gmail.com (S.I.); belen.montanel@hotmail.com (B.A.); eliecer.coto@sespa.es (E.C.)
2   Instituto de Investigación Sanitaria del Principado de Asturias—ISPA, 33011 Oviedo, Spain; lorcarebeca@gmail.com (R.L.); beadimo@gmail.com (B.D.-M.); josejucasa@yahoo.es (J.R.R.)
3   Redes de Investigación Cooperativa Orientadas a Resultados en Salud (RICORs), 28029 Madrid, Spain
4   Cardiología, HUCA, 33011 Oviedo, Spain; joseluis.lambert@gmail.com (J.L.L.-R.); ajuncovicente@gmail.com (A.J.-V.); vanesa.alonso@sespa.es (V.A.)
5   CSUR Cardiopatías Familiares HUCA, 33011 Oviedo, Spain
6   CIBER-Enfermedades Respiratorias, 28029 Madrid, Spain
*   Correspondence: juan.gomezde@sespa.es

**Abstract:** In this study we performed a next generation sequencing of 210 genes in 140 patients with cardiac failure requiring a heart transplantation. We identified a total of 48 candidate variants in 47 patients. Forty-three patients (90%) presented a single variant, and fourpatients (10%) were carriers of two variants. After refining the classification, we identified a pathogenic or likely pathogenic variant in 13 patients (10% of our cohort). In 34 additional cases (25%) the variants were classified as of unknown significance (VUS). In reference to the cause of cardiac failure in the 13 carriers of pathogenic variants, 5 were of dilated non-ischemic cause, 4 hypertrophic and 1 restrictive cardiomyopathy. In the ischemic cases (*n* = 3) no family history of cardiac disease was recorded, while nineof the non-ischemic had other relatives who were also diagnosed. In conclusion, the NGS of a cardiac transplanted cohort identified a definite or very likely genetic cause in 10% of the cases. Most of them had a family history of cardiac disease, and were thus previously studied as part of a routine screening by a genetic counselor. Pathogenic variants in cases without a family history of cardiac disease were mainly of ischemic origin.

**Keywords:** heart failure; heart transplant; familial cardiomyopathy; genetic screening

## 1. Introduction

Heart failure (HF) is the most severe manifestation of cardiac disease with increasing incidence among young adults in recent years. It has a heterogeneous cause with inherited and acquired risk factors and can be primary classified as of ischemic and non-ischemic origin.

Dilated cardiomyopathy (DCM) is the leading condition for heart failure. In some patients DCM is frequently due to mutations in the genes that encode proteins from the contractilemachinery. Other patients develop DCM as a secondary response to the limited blood supply to the cardiac myocytes, for instance, due to the ischemic coronary vessels. In addition to DCM, other cardiomyopathies related with mechanical or electrical dysfunction might result in HF, such as Hypertrophic (HCM), restrictive (RCM), left-ventricular non-compacted (LVNC) cardiomyopathies, among others [1]. In some patients the disease progresses toward a terminal state without pharmacological or surgical treatment to recover the contractile capacity, and the patients require a heart transplant [2]. DCM is the leading cause of heart transplantation (HT). According to the Spanish Cardiac Transplant Registry,

approximately 40% and 30% of the patients have DCM of non-ischemic and ischemic etiology, respectively, with 30% showing a different cause [3].

On the other hand, the role of inherited risk factors has anextreme manifestation in families with Mendelian forms of inherited cardiomyopathies [4,5]. However, ahigh number ofHF patients lack a clear family history of cardiac disease. Before the advent of the next-generation DNA sequencing techniques the search for mutations was based on the Sanger sequencing of candidate genes, a technique too costly to permitthe study of all HF patients. In addition, the number of candidate genes for cardiomyopathies have increased and some patients may harbor pathogenic variants in more than one gene. The interpretation of the disease candidate variants could thus be hampered by the lack of other recognized affected members of the family, who are necessary to confirm the segregation of the variant with the disease. Many labs limit the genetic study to patients with a clear family history of the disease, and the genetic cause of HF of any cause has been poorly understood in patients that do not meet the diagnostic criteria, such as elderly sporadic cases [6].

Therefore, the development of next-generation sequencing (NGS) techniques has facilitated the study of many genes at a minimum cost and labor requirements. The European Society of Cardiology recommends the use of NGS with panels formed by a large number of genes only when the family structure allows an analysis of the segregation of the candidate variant with the disease. However, less than 25% of the transplanted patients would have a clear family history and it would be important to validate the usefulness of genetic testing in these patients to characterize the genetic cause of HF and perform genetic counseling withtheir relatives [7,8]. The aim of this study was to characterize the genetic basis of HF in a cohort of cardiac transplanted patients.

## 2. Materials and Methods

### 2.1. Patients

We studied 140 patients who underwent a cardiac transplant in the period 2003–2018. They were recruited through the Cardiology Department of Hospital Universitario Central Asturias (HUCA), the reference center for this surgical procedure in the region. Only 10 of them (8%) were previously referred for the genetic screening forcardiovascular diseases. The study was approved by the HUCA Ethical Committee, and all the patients signed the informed consent for the genetic study. The mean age was 55 year ($\pm$9.7, range 17–70 years) and 104 (74%) were male. They were of European ancestry and from the region of Asturias, Northern Spain (total population approx. 1 million). In order to simplify the HF etiologies, we divided the patients intofourgroups: (1) ischemic cardiomyopathy (when the coronary disease is sufficient to explain the cardiac dysfunction); (2) inherited cardiomyopathies (HCM burn-out or RCM); (3) other causes (including drug-induced cardiomyopathy, myocarditis or valvular); (4) group of unknown etiology, considered "DCM non-ischemic cardiomyopathy". Table 1 summarizes the main characteristic of these patients.

### 2.2. Genetic Study and Variant Classification

We obtained the DNA from blood leukocytes of all the patients and performed the NGS of a total of 210 genes that have been associated with cardiovascular disease (Supplementary Table S1). These genes were sequenced with the Ion Torrent technology that uses semiconductor chips and the Ion GeneStudio S5 Sequencer (ThermoFisher Scientific, Waltham, MA, USA). The detailed procedure was previously reported [9,10]. The raw data was processed with the Torrent Suite v5 software. Reads assembling and variant identification were performed with the Variant Caller (VC). The Ion Reporter (ThermoFisher Scientific) and HD Genome One (DREAMgenics S.L., Oviedo, Asturias, Spain) software were used for variant annotation, including population, functional, disease-related and in silico predictive algorithms. The Integrative Genome Viewer (IGV, Broad Institute, Cambridge, MA, USA) was used for the analysis of depth coverage, sequence quality and

variant identification. We selected candidate variants based on both gene-associated to cardiomyopathies and/or aortopathies and frequency ≤5 carriers in the gnomAD database, according to the dominant pattern heritage of these diseases. We performed this filter in order to select variants which might be associated with heart failure transplantation, avoiding confusion with variants of other pathologies. All the variants classified as pathogenic or likely pathogenic were confirmed by Sanger sequencing of the corresponding PCR fragments (Supplementary Figure S1).

**Table 1.** Main characteristics of the 140 patients. The filtered variants identified in the primary analysis (*n* = 48) were found in 47 patients, with 4 cases harboring 2 variants.

| | Total N = 140 | DCM Non-Ischemic N = 52 | Ischemic N = 63 | HCM N = 9 | RCM N = 1 | Other N = 15 |
|---|---|---|---|---|---|---|
| Mean age ± SD | 55 ± 9.7 | 54 ± 10.5 | 55 ± 7.6 | 52 ± 12.2 | | 55 ± 14.8 |
| Range | 17–70 | 26–66 | 34–67 | 28–65 | | 17–70 |
| Male | 104 (74%) | 32 (62%) | 54 (86%) | 3 (33%) | | 14 (93.3%) |
| Cases with filtered variant (*n* = 47) | 47 | 16 | 19 | 5 | 1 | 6 |
| Carriers of pathogenic/Likely pathogenic variants (*n* = 13) | 13 | 5 | 3 | 4 | 1 | 0 |
| VUS carriers (*n* = 34) | 34 | 11 | 16 | 1 | 0 | 6 |

Based on the American College of Medical Genetics and Genomics (ACMG-AMP) criteria, candidate variants were classified as pathogenic/likely pathogenic, variants of uncertain significance (VUS) or likely non-pathogenic. The reference transcripts for the genes in which candidate variants were identified are presented as Supplementary Table S2.

## 3. Results

### 3.1. Genetic Characterization of the Study Cohort

Among the 140 patients requiring HT, 63 (45%) had an ischemic etiology (Supplementary Table S3), while 52 (37%) had DCM non-ischemic, 9 (6%) had HCM, 1 (1%) had RCM, and 15 (11%) had another cause (valvular heart disease, myocarditis, anthracycline and alcohol-induced cardiomyopathy).

We identified a total of 48 candidate variants in 47 patients by NGS. Forty-three patients (90%) presented a single variant, and fourpatients (10%) were carriers of two variants. We classified these variants according to the American College of Medical Genetics and Genomics guidelines (ACMG). Thus, 12 variants in 13 patients (LMNA p.Arg190Trp was found in two patients) were classified as likely pathogenic/pathogenic variants (10% of total cohort) (Table 2), and 36 variants in 34 patients were classified as variants of unknown significance (VUS) (25% of total cohort) (Table 3).

In the 10 patients with previous genetic screening, 6 of them harbored variants previously identified. In reference to the cause of HT in the 13 carriers of the 12 likely pathogenic/pathogenic candidate variants, 10 of them were due to non-ischemic causes, with 5 of them due to DCM, 4 HCM and 1 restrictive cardiomyopathy. On the other hand, threepatients suffered HT due to ischemic dilated cardiomyopathy, according to the recorded data of the heart transplantation unit. One patient harbored c.40406delC and c.48914_48915delTA truncating variants in the A-band of TTN gene, commonly associated with dilated cardiomyopathy (p.Pro13469GlnfsTer19 and p.Ile16305ArgfsTer6, respectively; NM_003319). Another one harbored the TNNT2 truncating variant c.823C>T, a gene associated withseveral cardiomyopathies, including HCM and DCM (p.Arg275Ter;

NM_001276345). Unfortunately, two of them were exitus at the time of the study, and one of them was not available. Thus, we cannot perform family studies in these patients.

**Table 2.** Pathogenic/likely pathogenic variants and associated heart transplantation cause. Variants were classified according to the consensus recommendation of the American College of Medical Genetics and Genomics and the Association for Molecular Pathology (Richards S, et al. Genet Med. 2015;17:405–424).

| Patient ID | Gene | cDNA | Exon | Effect | RefSeq Transcript | Cause | Age | Sex | Family Study |
|---|---|---|---|---|---|---|---|---|---|
| 1 | *FLNC* | c.322G>T | 1 | p.E108X | NM_001458 | DCM non-ischemic | 30 | Male | No |
| 2 | *LMNA* | c.568C>T | 3 | p.R190W | NM_170707 | DCM non-ischemic | 38 | Male | Yes |
| 3 | *LMNA* | c.568C>T | 3 | p.R190W | NM_170707 | DCM non-ischemic | 46 | Female | Yes |
| 4 | *LMNA* | c.481G>A | 2 | p.E161K | NM_170707 | DCM non-ischemic | 46 | Female | No |
| 5 | *MYBPC3* | c.2308+1G>A | IVS23 | splicing | NM_000256 | HCM | 63 | Male | Yes |
| 6 | *MYH7* | c.2464A>G | 22 | p.M822V | NM_000257 | HCM | 28 | Female | No |
| 7 | *MYH7* | c.1987C>T | 18 | p.R663C | NM_000257 | HCM | 43 | Male | Yes |
| 8 | *MYH7* | c.1208G>A | 13 | p.R403Q | NM_000257 | HCM | 48 | Female | Yes |
| 9 | *TNNT2* | c.823C>T | 16 | p.R275X | NM_001276345 | Ischemic | 48 | Male | No |
| 10 | *TNNT2* | c.516_518delGGA | 12 | p.E173del | NM_001276345 | Restrictive CM | 57 | Female | Yes |
| 11 | *TTN* | c.40406delC14 | 147 | p.P13469Qfs*19_delC | NM_003319 | Ischemic | 67 | Male | No |
| 12 | *TTN* | c.48914_48915delTA | 154 | p.I16305Rfs*6 | NM_003319 | Ischemic | 53 | Male | No |
| 13 | *TTN* | c.29453-1G>A | IVS118 | splicing | NM_003319 | DCM non-ischemic | 53 | Male | Yes |

**Table 3.** Variants of uncertain significance (VUS).

| Patient ID | Gene | cDNA | Exon | Effect | RefSeq Transcript | Cause |
|---|---|---|---|---|---|---|
| 14 | *TNNT2* | c.815A>G | 16 | p.N272S | NM_001276345 | Other |
| 15 | *RBM20* | c.1867C>T | 8 | p.R623W | NM_001134363 | Ischemic |
| 16 | *MYBPC3* | c.1025T>A | 12 | p.V342D | NM_000256 | Other |
| 17 | *LAMA4* | c.3059A>G | 23 | p.N1020S | NM_001105206 | Other |
| 18 | *FLNC* | c.1261C>T | 8 | p.R421W | NM_001458 | Ischemic |
| 19 | *MYBPC3* | c.1025T>A | 12 | p.V342D | NM_000256 | Ischemic |
| 20 | *MYH7* | c.1462T>A | 15 | p.F488L | NM_000257 | HCM |
| 21 | *DSP* | c.3919G>T | 23 | p.A1307S | NM_004415 | Ischemic |
| 22 | *TTN* | c.8342G>A | 35 | p.W2781X | NM_003319 | Ischemic |
| 23 | *TNNI3* | c.401A>T | 7 | p.D134V | NM_00363 | DCM non-ischemic |
| 24 | *JPH2* | c.1736C>T | 4 | p.P579L | NM_020433 | Ischemic |
| 25 | *FBN1* | c.5123G>A | 42 | p.G1708E | NM_000138 | Ischemic |
| 26 | *FLNC* | c.1216G>A | 8 | p.G406S | NM_001458 | DCM non-ischemic |
| 27 | *MYBPC3* | c.223G>A | 2 | p.D75N | NM_000256 | Ischemic |
| 28 | *FBN1* | c.5123G>A | 42 | p.G1708E | NM_000138 | Ischemic |
| | *MYBPC3* | c.1828G>C | 18 | p.D610H | NM_000256 | Ischemic |

**Table 3.** *Cont.*

| Patient ID | Gene | cDNA | Exon | Effect | RefSeq Transcript | Cause |
|---|---|---|---|---|---|---|
| 29 | *FBN1* | c.6401C>G | 53 | p.P2134R | NM_000138 | DCM non-ischemic |
| | *TTN* | c.1999C>T | 13 | p.R667X | NM_003319 | DCM non-ischemic |
| 30 | *MYH11* | c.3936G>C | 30 | p.K1312N | NM_001040114 | DCM non-ischemic |
| 31 | *TTN* | c.2703+1G>A | IVS16 | splicing | NM_003319 | Ischemic |
| 32 | *PKP2* | c.1988C>T | 10 | p.P663L | NM_004572 | Other |
| 33 | *JUP* | c.1000G>A | 6 | p.V334M | NM_002230 | Ischemic |
| 34 | *LDB3* | c.1165G>A | 8 | p.A389T | NM_007078 | Other |
| 35 | *BAG3* | c.1429C>T | 4 | p.R477C | NM_004281 | DCM non-ischemic |
| 36 | *JUP* | c.671A>T | 4 | p.K224M | NM_002230 | Ischemic |
| | *RYR2* | c.8407C>T | 56 | p.R2803W | NM_001035 | Ischemic |
| 37 | *TMEM43* | c.1178G>A | 12 | p.R393Q | NM_024334 | Ischemic |
| 38 | *MYPN* | c.3021A>C | 15 | p.E1007D | NM_001256267 | DCM non-ischemic |
| 39 | *DSP* | c.314G>A | 3 | p.R105Q | NM_004415 | Ischemic |
| | *LDB3* | c.324C>A | 4 | p.D108E | NM_007078 | Ischemic |
| 40 | *RBM20* | c.1275+2T>A | IVS2 | splicing | NM_001134363 | DCM non-ischemic |
| 41 | *TNNT2* | c.97G>A | 5 | p.E33K | NM_001276345 | DCM non-ischemic |
| 42 | *MIB1* | c.838_841delACTA | 6 | p.T280Qfs*15 | NM_020774 | Ischemic |
| 43 | *TCAP* | c.472C>A | 2 | p.R158S | NM_003673 | DCM non-ischemic |
| 44 | *MYPN* | c.3442G>A | 18 | p.A1148T | NM_001256267 | DCM non-ischemic |
| 45 | *TCAP* | c.16C>A | 1 | p.L6M | NM_003673 | Ischemic |
| 46 | *APOB* | c.4672A>G | 26 | p.T1558A | NM_000384 | DCM non-ischemic |
| 47 | *PKP2* | c.2502C>G | 13 | p.N834K | NM_004572 | Other |

*3.2. Family Studies*

We could perform family studies in 7 of the 13 carriers of pathogenic/likely pathogenic variant, four of them HCM, and three DCM non-ischemic (Supplementary Table S4). In the seven families, we screened 43 individuals (including the index cases) and 25 were carriers of the pathogenic/likely pathogenic variant. In 16 of these (64%) the disease was confirmed by clinical and/or imaging techniques. Interestingly, most of the asymptomaticcarriers were from a family with the c.29453-1G>A variant in the TTN gene, a putative splicing change. In this family, only the index case was affected while the six familial-carriers were unaffected. Thus, this variant could be re-classified as of uncertain significance (VUS).

**4. Discussion**

The genetic characterization of cardiac transplanted has been previously afforded by other authors [6,7,11]. In a study involving 26 Spanish patients, 50% showed a pathogenic mutation [6]. As expected, a family history was significantly higher among familial cases

compared to apparently sporadic (85% vs. 46%). Compared to this study our cohort has a much lower frequency of familial cases, that might explain the much lower frequency of patients with a pathogenic variant in our study. In addition, the mean age of our patients was 55 years compared to approximately 40 years in the Cuenca et al. cohort. Because it could be more probable to find a pathogenic variant among younger patients, the higher age of our patients might explain in part the lower genetic yield in our cohort.

It is possible that our study underestimates the rate of patients with a genetic cause of HF. This would be the case for some of the variants classified as VUS, in which familial segregation or functional studies should be necessary to refine the classification as pathogenic. Several patients also have more than one VUS or non-classifiable variant. It has been reported that compound heterozygosity of variants with reduced penetrance might increase the risk for several cardiopathies [11–13]. In these patients the relatives carrying a single mutation could lack clinical symptoms of cardiac disease, thus reducing the chance of familial classification. For instance, double heterozygosity for LMNA and TTN has been associated with a more severe clinical course of disease, in terms of age of terminal heart failure and heart transplant in a DCM family [14]. In these studies, the combination of multiple variants in the same individual caused earlier onset and more severe disease, although it is not known if or how this may modify the phenotype. In our study, for those patients with VUS a screening of the asymptomatic relatives should be necessary to uncover the segregation of the variants with the disease. We recognize the lack of this data as a limitation of our study, in which the classification of familial disease was based on the presence of symptomatic relatives.

We found three patients with LMNA variants. The LMNA gene encodes two intermediate filament proteins expressed in most differentiated somatic cells. These proteins form type A nuclear lamins and are involved in cellular and nuclear integrity and in the regulation of gene signaling and expression. The mutations in this gene cause a variety of laminopathies, selectively affecting different tissues and organ systems. However, the most common laminopathy is that which affects the heart and causes dilated cardiomyopathy, with or without skeletal muscle involvement [15]. By the age of 60, 55% of LMNA gene mutation carriers die of cardiovascular death or receive a heart transplant, compared with 11% of patients with idiopathic cardiomyopathy without LMNA mutation [16]. In another study, it has been seen that 20% of patients with a mutation in the LMNA gene required a heart transplant. These patients would have a severe and progressive phenotype, as well as a poor prognosis of the disease, such assuffering sudden cardiac death as the first symptom of the disease. This study highlights the importance of early family screening in young family members and the clinical follow-up of patients with a positive LMNA genotype to provide preventive treatment [17]. Therefore, benefits of genetic screening are far beyond possible long-term ethical issues, such as psychological distress. However, due to the potential ethical considerations of these studies, all patients who underwent genetic screening signed an informed consent which stated if they wanted to know the result of the genetic test.

Family history is essential to determine a possible genetic cause of dilated cardiomyopathy. In a study of patients with familial dilated cardiomyopathy and conduction block, 19.5% were found to have LMNA mutations [18]. Therefore, the presence of premature conduction system disease in combination with unexplained dilated cardiomyopathy should lead cardiologists to seriously consider the LMNA mutation as a cause.

In our cohort, two patients were identified as carriers of a variant in the 190 position.

The codon 190, located in the rod-domain lamin A/C, is the mutational hotspot. This would lead to molecular changes in the lamin A/C structure that can decrease the mechanical stability of the muscle, which is critical during muscle contraction. Several mutations were described at this position, most associated with DCM: p.Arg190Gln, p.Arg190Trp, p.Arg190fsX22 and p.Arg190Pro [19].

On the other hand, we have found threevariants (TNNT2 p.R275X, TTN p.P13469Qfs*19 and p.I16305Rfs*6) classified as pathogenic/likely pathogenic in genes associated with car-

diomyopathies and heart disease, in patients whose reason for HT was due to an ischemic cause. Both ischemic dilated cardiomyopathy and non-ischemic dilated cardiomyopathy are characterized by progressive contractile dysfunction leading to left ventricular dilation and heart failure [20]. Moreover, in somecases genetic variants may play a key role in the susceptibility to DCM by enhancing the phenotype, as well as other environmental triggers such as atrial fibrillation or alcohol. For instance, pathogenic variants in TTN more frequently presented with severe DCM (ejection fraction ≤20%), which improves with treatment [21]. Therefore, we highlight the importance of cardiomyopathy-associated genetic variants. In addition, only 8% of the patients had been previously genetically screened. We identified a pathogenic/likely pathogenic variant in 10% of the patients, and a VUS variant in 25% of the patients, which could increase the genetic yield of our study. Thus, more than 8% of these patients should have been referred forgenetic screening. This could be because our heart transplantation cohort began in 2003, and many of these patients were directly referred to heart failure units.

### 5. Conclusions

We have identified a significant percentage of new or rare genetic variants in genes that would be associated with heart diseases in a heart transplanted cohort. Our results suggest the importance of retrospective genetic studies in a cohort of ancient cases of heart transplant patients that had not been studied before. In addition, it would be important to reevaluate the reason for transplantation, since they can be initially diagnosed due to ischemic causes, but eventuallypathogenic/likely pathogenic variants are found, so it should be reviewed in their family. Detection of these variants might be helpful to achieve an early diagnosis of these diseases.

**Supplementary Materials:** The following supporting information can be downloaded at: https://www.mdpi.com/article/10.3390/cardiogenetics12020018/s1, Figure S1: IGV and Sanger sequence of MYH7 p.R403Q (c.1208 G>A) in reverse strand.; Table S1: Genes included in the NGS; Table S2: Transcripts for the genes with candidate variants; Table S3: Main clinical characteristics of the ischaemic cases.; Table S4: Family studies in available patients.

**Author Contributions:** Conceptualization, R.L., J.R.R. and J.G.; formal analysis, E.C.-L.; investigation, E.C.-L., R.L., B.D.-M., J.L.L.-R., J.R.R., S.I., B.A., A.J.-V. and V.A.; data curation, E.C.-L.; writing—Original draft preparation, E.C.-L. and E.C.; writing—Review and editing, R.L.; supervision, E.C. and J.G.; project administration, J.G.; funding acquisition, J.G. All authors have read and agreed to the published version of the manuscript.

**Funding:** This research was funded by Instituto de Salud Carlos III (ISCIII), grant number PI17/00648.

**Institutional Review Board Statement:** The study was conducted in accordance with the Declaration of Helsinki, and approved by the Institutional Review Board of Hospital Universitario Central de Asturias. All participants signed and informed consent to give up samples for research studies.

**Informed Consent Statement:** Informed consent was obtained from all subjects involved in the study.

**Data Availability Statement:** Further data is available by mailing to the corresponding author (juan.gomezde@sespa.es).

**Acknowledgments:** Authors want to acknowledge to all participants in the study and their families. This work was supported by a grant from the Spanish Plan Nacional de I + D + I Ministerio de Economía y Competitividad and the European FEDER, grant ISCIII PI17/00648 (JG).

**Conflicts of Interest:** Authors declare no conflict of interest.

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
