# Peer review of "Genetic Screening of a Large Panel of Genes Associated with Cardiac Disease in a Spanish Heart Transplanted Cohort"

_cardiogenetics, doi:10.3390/cardiogenetics12020018_

Round 1
Reviewer 1 Report
The authors contribute with genetic data regarding patients who progress to terminal HF, requiring transplantation. An interesting result was obtained in patients with ischemic heart disease, which highlights the limitations of clinical classifications that may underestimate the role of genetics in HF pathophysiology. This reinforces the need for further studies including patients with different HF etiologies to understand the overall contribution of cardiomyopathies-related genes.
Reviewer 2 Report
This study investigates yield of genetic screening with a large NGS panel in heart transplantation recipients. The topic is interesting, and in particular the fact that Authors included in the analysis all aetiologies of HF. However, I have several concerns:
1) As said, the fact that both patients with primary cardiomyopathies and with secondary causes of HF (i.e., ischaemic) were included in the analysis is interesting, but needs refinement. In particular:
- a definition for ischeamic dilated cardiomyopathy must be provided. In this regard, may some cases of 'ischaemic HF' carriers of variants in DCM-genes be reclassified? Sometimes a dilated heart is labelled as 'ischaemic HF' even if with only a single coronary vessel disease, which is not correct. Did all the 'ischaemic patients' had consistent diagnosis?
- this aspect may deserve a wider consideration in Discussion. If all cases of 'ischaemic' HF are 'true', what is the role of cardiomyopathy-variants in their clinical course?
- Do you have available information also about family screening in 'ischaemic' patients with positive genetics? This would significantly improve the results of the paper.
- at least once in the manuscript please specify what are all the 'other' causes of HF.
2) I would suggest to confine the genetic screening to variants associated with cardiomyopathies and exclude those associated with other diseases such as variants in LDL receptor.
3) In Results, line 107: "We selected candi-107 date variants based on both gene-associated to cardiomyopathies and/or aortopathies and 108 frequency ≤5 carriers in the gnomAD database, according to dominant pattern heritage of 109 these diseases". This should be included in Methods and better explained because it is not clear.
4) Introduction is very redundant and should be comprehensively rephrased.
Reviewer 3 Report
In the study by Elias Cuesta-llavona et al., they investigated Genetic screening of a large panel of genes associated with cardiac disease in a Spanish heart transplanted cohort.
Although there are many reports of genetic abnormalities involved in the occurrence of cardiomyopathy, there are only a limited number of reports of genetic abnormalities involved in the development and aggravation, and comprehensive research is desired. However, it has not been done yet.
Although the number of cases in this study remains small, the significance of reporting is high from the point of view.
In addition, this study is very unique because it focuses on not only etiology of cardiomyopathy but also ischemic heart disease notably. However, this study has some limitations and should be addressed appropriately.
Not all patients with heart failure including cardiomyopathy lead to a heart transplant, but some of them. In other words, it is important to identify the genetic abnormality that causes aggravation of heart failure. Although there are reports that familial cardiomyopathy and cardiomyopathy with genetic abnormalities tend to become more severe, these relationship is still controversial. Empirically, we met some heart transplant patients who died or even had a heart transplant in their family or siblings.
Do you have such patients in your 140 database?
Has the gene variant been found in those patients?
Please investigate the differences in clinical background between the 48 patients in whom the genetic variant was found and those in other patients, e.g. etiology, comorbidity and especially family history of severe heart failure and sudden death.
Too much difference in trends by each surveys symbolizes the complexity and difficulty of expanding gene variant research. According to the data in your suppl table, only patients 4, 6, 8 and 11 have the same genetic variant in the family, and the others appear to be isolated. I feer your results also very vague, as previous studies. For example, in your study, there are only three LMNA-related gene variant that are serious and important variant so far. In addition, patient Nr 3 who underwent a family survey appears to be isolated. Is families of patient Nr 3 safe? How dangerous is the career without myopathy of families of patient Nr 4 ? Identifying the LMNA gene can occures ethical issues if it can reduce families' quality of life.
The discussion on the ethical issues of genetic research is inadequate. Discussing what causes this ambiguity and focusing on this issue is crucial to the clinical significance of future genetic analysis.
It is very unique to identify the genes involved in the heart transplantation even in ischemic cardiomyopathy.
The number of cases is extremely small, but it should be emphasized more.
We have experienced many patients who developed severe cardiac remodeling even with a clinically minor history of ischemia.
Therefore, the treatment history of these patients is very important.
Please investigate the medical history and treatment history of ischemic cardiomyopathy. Presence or absence of old myocardial infarction and ischemic area, history of CABG or PCI treatment, presence or absence of non-reperfusion blood vessels, etc.
Round 2
Reviewer 2 Report
Revision improved the manuscript, and in particular the Authors are to be praised for the meticulous revision of cases as asked by reviewers.
However, the paper still needs ameliorations, but mostly in terms of adequateness of presentation:
1) very good for the Authors to have double checked on 'ischaemic' HF. However, it is not necessary to report this in the paper and I suggest to erase all the "We former classified 72 patients as ischemic cases based on the reason of trasplantation written down in the clinical history. In order to perform a consistent diagnosis, we revised these 72 cases, thus reclassified 9 of them to non-ischemic groups, including one who harbor a likely pathogenic variant (patient 4), which highlights the limitations of clinical classifications that may underestimate the role of genetics in HF pathophysiology. " part.
On the contrary, it is of paramount importance to report the specific definition of each HF condition in the Methods, and in particular of 'ischaemic DCM'. This should not only be that of coronary disease not able to explain the dysfunction, but more specific (i.e., haemodinamically significant three vessel disease...).
2) I do not agree with all this part of Discussion: "Therefore, these carriers with an ischemic cause could be due to an initial misclassification, being finally due to an ischemic cause associated with a DCM. Heart failure (HF) is not a single pathological diagnosis, but a clinical syndrome consisting of cardinal symptoms that may be accompanied by clinical signs. HF is due to a structural and/or functional abnormality of the heart that results in elevated intracardiac pressures and/or inadequate cardiac output at rest and/or during exercise (20). Identification of the etiology of the underlying cardiac dysfunction is of almost importance as the specific underlying pathology can determine a specific treatment. The most common causes are cardiovascular disease (ischemic cardiomyopathy, when the coronary disease is sufficient to explain the cardiac dysfunction), hypertension, valve disease, arrhythmias, inherited cardiomyopathies, infiltrative, congenital heart disease, infective and drug-induced cardiomyopathy. Moreover, there are other causes, such as, storage disorders, pericardial, metabolic, endomyocardial disease or neuromuscular disease (20). The term "ischemic cardiomyopathy" is used to describe a clinical syndrome similar to primary dilated cardiomyopathy, secondary to prolonged, severe, diffuse coronary artery disease, including acute infarction. Both ischemic cardiomyopathy and non-ischemic cardiomyopathy are characterized by progressive contractile dysfunction leading to left ventricular dilation and heart failure"
which i feel is not at all consistent with the message that the paper is trying to send. The point is not that ischaemic DCM patients with P/LP variants were misclassified, but that in some cases cardiomyopathy associated genetic variants may play a role in susceptibility to DCM in environmental setting such as in fact ischaemic disease. Please report on this aspect.
Reviewer 3 Report
This is a second round review of the manuscript, written by Elias Cuesta-llavona and colleagues, on the difficult topic of Relation between heart transplantation and genetic abnormalities. The comments send by the authors do not improve the manuscript in some parts of way suggested by the reviewer. Thus, the previous recommendation is maintained.
The author said that they have added a table with the main clinical characteristics of the ischemic cases in supplementary table 3,  however the new supplemental table 3 doesn't look much different from the old version to me. Please stated clearly a treatment history of ischemic heart disease in detail.
the author said that this study highlights the importance of early family screening in young family members and the clinical follow-up of patients with a positive LMNA genotype to provide preventive treatment. However I pointed out, not the importance of follow-up, but the ethical approach to the negative effects on ADL of patients with genetic variant.
It is clearly important to follow patients who have investigated the gene mutation, but do patients need to be informed of the gene mutation as an uncertain prognostic factor and cause long-term psychological distress? Without proceeding with this discussion, we cannot hope for the development of research on gene analysis in the future.
Long-term follow-up can also impose a financial burden for these patients. Please deepen your discussion on this ethical issue.
